# Short- and Long-Term Outcome of Laparoscopic- versus Robotic-Assisted Right Colectomy: A Systematic Review and Meta-Analysis

**DOI:** 10.3390/jcm11092387

**Published:** 2022-04-24

**Authors:** Peter Tschann, Philipp Szeverinski, Markus P. Weigl, Stephanie Rauch, Daniel Lechner, Stephanie Adler, Paolo N. C. Girotti, Patrick Clemens, Veronika Tschann, Jaroslav Presl, Philipp Schredl, Christof Mittermair, Tarkan Jäger, Klaus Emmanuel, Ingmar Königsrainer

**Affiliations:** 1Department of General and Thoracic Surgery, Academic Teaching Hospital Feldkirch, 6800 Feldkirch, Austria; markus.weigl@lkhz.at (M.P.W.); stephanie.rauch@lkhf.at (S.R.); daniel.lechner@lkhf.at (D.L.); stephanie.adler@lkhf.at (S.A.); paolo.girotti@lkhf.at (P.N.C.G.); ingmar.koenigsrainer@lkhf.at (I.K.); 2Institute of Medical Physics, Academic Teaching Hospital Feldkirch, 6800 Feldkirch, Austria; philipp.szeverinski@lkhf.at; 3Private University in the Principality of Liechtenstein, 9495 Triesen, Liechtenstein; 4Department of Radio-Oncology, Academic Teaching Hospital Feldkirch, 6800 Feldkirch, Austria; patrick.clemens@lkhf.at; 5Department of Internal Medicine II, Academic Teaching Hospital Feldkirch, 6800 Feldkirch, Austria; tschann.veronika@lkhf.at; 6Department of Surgery, Paracelsus Medical University, 5020 Salzburg, Austria; j.presl@salk.at (J.P.); p.schredl@salk.at (P.S.); ta.jaeger@salk.at (T.J.); k.emmanuel@salk.at (K.E.); 7Department of Surgery, St. John of God Hospital, Teaching Hospital of Paracelsus Medical University, 5020 Salzburg, Austria; christof.mittermair@bbsalz.at

**Keywords:** robotic surgery, laparoscopic surgery, right colectomy, short-term outcome, long-term outcome, costs

## Abstract

Background: There is a rapidly growing literature available on right hemicolectomy comparing the short- and long-term outcomes of robotic right colectomy (RRC) to that of laparoscopic right colectomy (LRC). The aim of this meta-analysis is to revise current comparative literature systematically. Methods: A systematic review of comparative studies published between 2000 to 2021 in PubMed, Scopus and Embase was performed. The primary endpoint was postoperative morbidity, mortality and long-term oncological results. Secondary endpoints consist of blood loss, conversion rates, complications, time to first flatus, hospital stay and incisional hernia rate. Results: 25 of 322 studies were considered for data extraction. A total of 16,099 individual patients who underwent RRC (*n* = 1842) or LRC (*n* = 14,257) between 2002 and 2020 were identified. Operative time was significantly shorter in the LRC group (LRC 165.31 min ± 43.08 vs. RRC 207.38 min ± 189.13, MD: −42.01 (95% CI: −51.06−32.96), *p <* 0.001). Blood loss was significantly lower in the RRC group (LRC 63.57 ± 35.21 vs. RRC 53.62 ± 34.02, MD: 10.03 (95% CI: 1.61–18.45), *p* = 0.02) as well as conversion rate (LRC 1155/11,629 vs. RRC 94/1534, OR: 1.65 (1.28–2.13), *p* < 0.001) and hospital stay (LRC 6.15 ± 31.77 vs. RRC 5.31 ± 1.65, MD: 0.84 (95% CI: 0.29–1.38), *p* = 0.003). Oncological long-term results did not differ between both groups. Conclusion: The advantages of robotic colorectal procedures were clearly demonstrated. RRC can be regarded as safe and feasible. Most of the included studies were retrospective with a limited level of evidence. Further randomized trials would be suitable.

## 1. Introduction

With the introduction of minimal invasive surgery in 1991 for colorectal diseases, a new era of surgery was established and increased rapidly. The benefits of minimal invasive surgery are clear and well-demonstrated in previous literature. Short-term benefits, such as more rapid postoperative recovery, less incisional trauma, less pain, faster intestinal passage, shorter hospital stay, are clearly demonstrated [1,2,3,4,5]. Moreover, minimal invasive surgery showed comparable oncological outcomes and increased short-term results compared to open surgery [2,4,6]. Robotic surgical systems have several technological advantages compared to conventional minimal invasive techniques, including a high-definition field of view, articulated instruments, tremor filtering and better ergonomics, which may translate intraoperative movements to a more precise dissection [7,8,9]. The disadvantages of robotic surgery include the lack of tactile feedback, additional surgery time and higher costs [7,9,10,11,12].

In minimal invasive surgery for right-sided colorectal resections, there is a growing literature comparing the outcome of robotic right colectomy (RRC) versus laparoscopic right colectomy (LRC) [7,8,10,13,14,15,16,17,18,19,20,21,22,23,24,25,26,27,28,29,30,31,32,33,34,35,36]. Most of them consist of retrospective, single-center experiences with small sample sizes, different techniques for colorectal cancer surgery and nearly no data on the oncological long-term outcome.

With this background and in order to highlight the benefits of robotic right colectomy regarding short- and long-term outcomes, we achieved this manuscript as a systematic review and a meta-analysis of literature which compares laparoscopic and robotic right-sided colorectal resections.

## 2. Methods

### 2.1. Literature Search Strategy and Eligibility Criteria

After an institutional review board approval, this study was conducted according the PRISMA Statements (updated 2020) for review and meta-analysis [37]. A systematic search in literature was performed in PubMed, Embase and Scopus databases for comparative studies published between 1 January 2000 and 30 September 2021. Search terms used were “laparoscopic versus robotic colectomy”, “laparoscopic versus robotic right colectomy”, “laparoscopic versus robotic CME”, laparoscopic versus robotic complete mesocolic excision”, “laparoscopic versus robotic right sided colorectal cancer”, “robotic [AND] laparoscopic right hemicolectomy” and “laparoscopic versus robotic right hemicolectomy”. Results from the databases were compared to sustain a single list of articles for screening. Titles, abstracts and full-text articles were screened and selected by two authors (PT and MW). Disagreement was addressed by discussion and followed by consensus. Duplicate references were removed by manual search.

Only full-text studies published in the English language which specifically compared elective laparoscopic versus robotic right colectomy were considered. Comparative studies with fewer than 15 participants, pediatric studies (age under 18) and studies which lacked a robotic group or vice versa were excluded. Each journal’s score (e.g., journal’s Impact Factor) of included manuscripts was not a factor of exclusion.

The primary endpoint was postoperative morbidity and mortality and long-term oncological results. Secondary endpoints consisted of blood loss, conversion rates, complications, time to first flatus, hospital stay and incisional hernia rate.

### 2.2. Assessment of Data Extraction and Methodological Quality

Data extraction included study characteristics (name of primary author, country of origin, study period and study design), patients’ characteristics (age, sex, body mass index (BMI), American Society of Anesthesiologists Score (ASA) [38]), intraoperative blood loss, type of anastomosis, operative time, conversion to open surgery and number of harvested lymph nodes and postoperative variables (hospital mortality, overall morbidity, anastomotic leak, postoperative hemorrhage, abdominal abscess, time to first flatus, postoperative ileus, wound infections, length of hospital stay, incisional hernia, quality of surgery, local recurrency and oncological 3 and 5 years disease free and overall survival rates).

The methodological quality was assessed by two authors independently (DL and SR). The MINORS scale [39] was used to evaluate the quality for cohort studies, while the Jadad scoring [40] was used for randomized controlled trials.

### 2.3. Statistical Analysis

For the continuous variables, the inverse variance method was applied, and the averaged means and standard deviations were reported. Comparisons were expressed as the mean difference (MD) and the 95% CI. If continuous data from individual studies were reported as the median (interquartile range, IQR, or range), then these data were transformed into mean and standard deviation suggested by Hozo et al. [41]. Costs were expressed in Euros by a factor of 1000. For dichotomous variables, the Mantel–Haenszel method was used, and comparisons were reported as odds ratios and the associated 95% CI. If no events occurred in either study arm, then these studies were excluded from the calculation. Funnel plots were also created for all calculations to look at potential publication bias. Heterogeneity was determined using the I^2^ statistic and interpreted as indicated in the Cochrane handbook for Systematic Reviews [42]. I^2^ values of 0% to 40% were interpreted as might not be important, 30% to 60% as may represent moderate heterogeneity, 50% to 90% as may represent substantial heterogeneity and 75% to 100% as considerable heterogeneity. For all calculations, the random-effects model was applied because of the expected heterogeneity among all included studies. A *p*-value of less than 0.05 was determined to be statistically significant. Statistics were performed using Review Manager software version 5.4.1 (Cochrane Collaboration, Copenhagen, Denmark).

## 3. Results

### 3.1. Search Results and Study Details

Database search and manual screening yielded a total of 322 potentially relevant studies (Figure 1).

Of these, 25 articles published between 2007 and 2021 were considered for data extraction and were included into this meta-analysis [7,8,10,11,13,14,15,16,17,18,19,20,21,22,23,24,25,26,27,28,30,31,32,33,34,35,36]. The majority of included trials were of a retrospective study design, 2 prospective studies [30,32] and 1 randomized controlled study [7,11] fulfilled criteria for data extraction. A total of 16,099 individual patients who underwent RRC (*n* = 1842) or LRC (*n* = 14,257) between 2002 and 2020 were identified. Study details and quality assessment of included studies are summarized in Table 1. Funnel plots did not show evidence of significant bias among the included studies.

### 3.2. Patients’ Characteristics

Details of included patients regarding age, sex, BMI, ASA > 2 and number of cancer cases of retrieved studies are shown in Table 2. Patients in the LRC group were significantly older than in the RRC group (LRC 70.27, ±3.00 vs. RRC 68.79, ±2.90, *p* = 0.03).

### 3.3. Perioperative Outcomes

Perioperative outcomes are shown in Table 3. Operative time, which could be extracted from 16 of 25 included studies, was found to be significantly shorter in the LRC group (LRC 165.31 min ± 43.08 vs. RRC 207.38 min ± 189.13, MD: −42.01 (95% CI: −51.06 −32.96), *p <* 0.001). Blood loss was significantly lower in the RRC group (LRC 63.57 ± 35.21 vs. RRC 53.62 ± 34.02, MD: 10.03 (95% CI: 1.61–18.45), *p* = 0.02) as well as conversion rate to an open procedure (LRC 1155/11629 vs. RRC 94/1534, OR: 1.53 (1.08–2.17), *p* = 0.02) and hospital stay (LRC 6.15 ± 31.77 vs. RRC 5.31 ± 1.65, MD: 0.84 (95% CI: 0.29–1.38), *p* = 0.003). Intracorporeal anastomosis procedures were significantly more often performed in the robotic group compared to the laparoscopic procedures (LCR 329/4308 vs. RRC 468/860, OR: 0.03 (0.00–0.20), *p* < 0.001). Mortality rate was low in both groups (LRC 126/13388 vs. RRC 18/1198, 0.66 (0.41–1.06), *p* = 0.08). Postoperative overall morbidity (LRC 3093/14242 vs. RRC 464/1825, OR: 1.01 (0.86–1.19), *p* = 0.88) did not differ between both procedures. Other complications such as anastomotic leakage, postoperative hemorrhage, postoperative ileus, wound infection, non-surgical complications and abdominal abscess did not differ as well between both groups.

### 3.4. Oncological Findings

Oncological findings are shown in Table 4. A total of 17 studies reported about the number of retrieved lymph nodes, which showed no significant difference between both groups (LRC 22.97 ± 5.94 vs. RRC 23.82 ± 6.76, MD: −0.85 (95% CI: −2.19–0.48), *p* = 0.21). Only 4 studies performed a long-term observation of their patients [7,8,11,13,33]. These showed no difference regarding 5-years disease free (LRC 178/213 vs. RRC 162/190, OR: 0.87 (0.50–1.51), *p* = 0.62) or overall survival (LRC 172/213 vs. RRC 157/190, OR: 0.90 [0.54–1.52], *p* = 0.7). Pathological TNM or UICC staging and information about adjuvant treatment of studies with data about oncological long-term follow up are shown in Appendix A. Pathological TNM staging was assessed by 18 authors [7,8,10,11,13,14,15,16,17,18,19,20,21,22,23,24,25,26,27,28,30,31,32,33,34,35,36] (Appendix A). UICC staging was included in 3 studies [8,10,23] (Appendix A).

### 3.5. Costs

Meta-analysis of surgery specific costs and total costs was evaluated by 4 authors [7,11,16,30,33] and is shown in Appendix A. Surgery specific costs (LRC 3.900 ± 1.677 vs. RRC 8.156 ± 0.458, MD: −4.16 (95% CI: −7.12–−1.21), *p*= 0.006) and total costs (LRC 7.647 ± 1.307 vs. RRC 10.306 ± 1.507, MD: −2.66 (95% CI: −5.17–−0.15), *p* = 0.04) were significantly higher in the RRC group.

## 4. Discussion

In this meta-analysis, we could clearly demonstrate that robotic right colectomy is safe and feasible regarding perioperative morbidity and mortality compared to conventional laparoscopic procedures. Surgical specific complications such as anastomotic leakage, postoperative bleeding, ileus and wound infection were similar between both groups. Including more than sixteen thousand patients in this review, our results confirmed some of that what previous studies already suggested in terms of perioperative findings [9,43,44,45]. Moreover, this is to our knowledge the first meta-analysis including long-term oncological results. Our findings showed no difference between both techniques and underline the oncological efficiency of robotic procedures.

Meta-analysis of studies comparing laparoscopic versus robotic rectal resection showed several advantages for the robotic technique: reduced estimated blood loss, lower intraoperative conversion rate and no difference regarding postoperative morbidity [46,47]. Some previously conducted meta-analyses for right colectomies found controversial findings. In particular, a lower overall complication rate for RRC was shown by three previous meta-analyses [12,44,48]. Regarding intraoperative blood loss, we could show a significant difference favoring the robotic group, whereas two studies [9,46] showed no difference between RRC and LRC. Consistent with our results, robotic surgery had a lower intraoperative blood loss in four other systematic reviews [12,44,48,49] and a lower conversion rate in three studies [9,44,49]. Only two meta-analyses reported a shorter length of hospital stay for the RRC group compared with LRC [44,49]. The operative time and surgery specific costs as well as total costs were significantly higher in the robotic group which is also described in several previous performed meta-analyses [9,12,44,46,48,49]. A reason for this could be that case complexity is increasing with the progress of performed cases [50]. In a recently published article by Nasseri et al., this assumption was confirmed also with a higher experience in robotic colorectal surgery [51]. Another reason could be that docking time and changing of instruments are more time consuming in robotic surgery. Regarding operative time, the role of the type of anastomosis—either extracorporeally or intracorporeally—is still discussed controversially. In a meta-analysis by Genova et al., it was shown that operative time was significantly shorter independent of the type of anastomosis in the laparoscopic group [44]. In the subgroup analysis comparing only extracorporeal anastomosis, the meta-analysis published by Solaini et al. found no difference between RRC and LRC [9]. However, most of the surgeons prefer to perform the intracorporeal anastomosis with the robotic system as it decreases the difficulty of intracorporeal suturing dramatically.

In this meta-analysis we could not show a significant difference concerning harvested lymph nodes between RRC and LRC. Contradictory to our results, Genova et al. showed a significantly higher number of harvested lymph nodes in the LRC group [44]. Solaini et al. reported a tendency toward a higher number of harvested lymph nodes during RRC and a significantly reduced conversion rate which may indicate the advantage of robotic surgery in performing tissue dissection [9]. Further interpretation about discrepancies of those results should be conducted with caution: most of the included studies were of a retrospective design. Information about the extension of lymphadenectomy is not provided in the majority of the studies even if a CME (complete mesocolic excision) is stated as the surgical standard procedure. Nevertheless, the advantage of RRC regarding soft tissue dissection is well-documented and may explain the tendency of our results toward a higher number of lymph nodes in RRC.

Only four studies reported a long-term oncological follow-up [7,8,13,33]. Those data showed no difference between both groups regarding disease free and overall survival. Pathological tumor stage did not differ between LRC and RRC in those studies (Appendix A). Only one study by Park et al. mentioned if adjuvant treatment was performed (Appendix A). However, the oncological long-term outcome should be interpreted with caution. Only one randomized controlled trial is reporting about long-term data with a relatively low number of patients, although it is important to mention that at the beginning of a robotic program, only specialized colorectal surgeons are performing the robotic cases. Therefore, the quality of tissue dissection is expected to be equal between both groups and can explain similar oncological short- and long-term outcomes.

In previous literature, several studies showed that minimal invasive colorectal surgery is associated with a better short-term outcome such as reduced postoperative pain, faster recovery and shorter hospital stay than open procedures [4,49]. We could show a significantly shorter length of stay in the RRC group. This could be explained because of an age difference between both groups. Patients in the LRC group were significantly older than in the RRC group (LRC 70.27, ±3.00 vs. RRC 68.79, ±2.90, *p* = 0.03). Patients’ selection of younger and healthier patients in favor of robotic procedures may indicate a selection bias of retrospective studies. Controversially to our results, Ma et al. showed an advantage for the LRC group [49], whereas three other meta-analyses showed no difference [9,44,48] regarding the length of hospital stay. This may be explained by including more recently published literature. Only looking at literature published 2018 or later, we observe a similar or shorter length of stay in those studies [7,8,14,16,18,20,22,23,25,28,35]. Only one study showed contrary findings in a recently published retrospective trial [21].

Perioperative findings such as time to first flatus and overall morbidity showed no difference between both groups and confirmed data from previous published meta-analyses [9,43,44,45]. A lower overall complication rate for RRC was reported by three meta-analyses [12,44,48]. However, the differences between RRC and LRC especially in terms of morbidity are hardly visible. Compared to open surgery, RRC is to be shown favorable in terms of complications and surgical side infections in the work by Widmar et al. [19], but data are limited that only one study also included open procedures.

Consistent with previous published meta-analyses [44], our data showed a significantly higher rate of intracorporeal performed anastomosis in the robotic group compared to conventional laparoscopic surgery (LRC 329/430 (7.6%), RRC 468/860 (54.4%), OR: 0.03 (0.00–0.20), *p* < 0.0001), perhaps because of different levels of technical difficulty of both techniques. However, in robotic surgery, intracorporeal suturing is easy to perform, comfortable and safe, whereas in laparoscopic surgery, extracorporeal anastomosis is favored. Moreover, consistent with a propensity-matched comparison of 379 intracorporeal anastomosis procedures (335 robotic and 44 laparoscopic), robotic surgery showed a significantly lower conversion rate, shorter length of stay and fewer postoperative complications [52].

To date, only four studies [7,8,13,33] compared 5-years oncological data, and only one of them was of a randomized study design [7]. However, these data showed that robotic surgery is safe concerning the oncological outcome and number of harvested lymph nodes. These data are limited because of a limited number of patients followed-up in a randomized controlled trial by Park et al. [7].

This meta-analysis presents a few limitations. First, most of the included studies were of a retrospective study design, and only one randomized trial was found in literature [7,11]. The risk of important bias is relevant. The current literature lacks randomized controlled trials or studies of higher quality. Second, most of the papers described differences regarding the technique of anastomosis and no specification about tumor localization which could have a bias on outcome in both groups. Data heterogeneity within this study was often high because of the retrospective study design of included studies. No detailed information was available concerning the method for measuring several outcomes. Indication for surgery, histopathological work-up (Appendix A) and information about adjuvant treatment (Appendix A) were not provided in the majority of included studies. Third, the role of a learning curve on perioperative findings, postoperative outcome and costs is still not demonstrated yet. Only 11% of all included patients were treated robotically. With a higher number of robotic cases, the learning curve may be completed, and operative time could be lower in the robotic group. However, trocar placement and docking of the robotic system will require time also in case of a completed learning curve. Fourth, pathological results are not quoted in the majority of the included studies. The indication for surgery and complexity of the cases in both groups could not be identified and may implicate a selection bias especially in non-randomized trials. Finally, only four studies reported about oncological long-term data. Furthermore, description about the extension of the lymph node field or CME is hardly available in current literature, and a conclusion about the follow-up should be conducted carefully. Further studies are needed to verify the tendency of a similar oncological outcome in this study.

However, based on the results, we found in this meta-analysis that robotic right colectomy is safe and feasible regarding perioperative findings, postoperative outcome and oncological long-term data—so far available—compared to conventional laparoscopic procedures.

## 5. Conclusions

This meta-analysis shows that robotic right colectomy is advantageous over laparoscopic procedures regarding intraoperative blood loss, conversion rate to open procedures and in terms of length of hospital stay. Other clinical findings or oncological outcome appear to be equivalent between both groups. However, RRC can be regarded as a safe and feasible technique for right-sided colectomy. Further prospective randomized conducted trials with a long-term follow-up would be suitable to achieve a higher level of evidence especially regarding oncological long-term outcome.

## Figures and Tables

**Figure 1 jcm-11-02387-f001:**
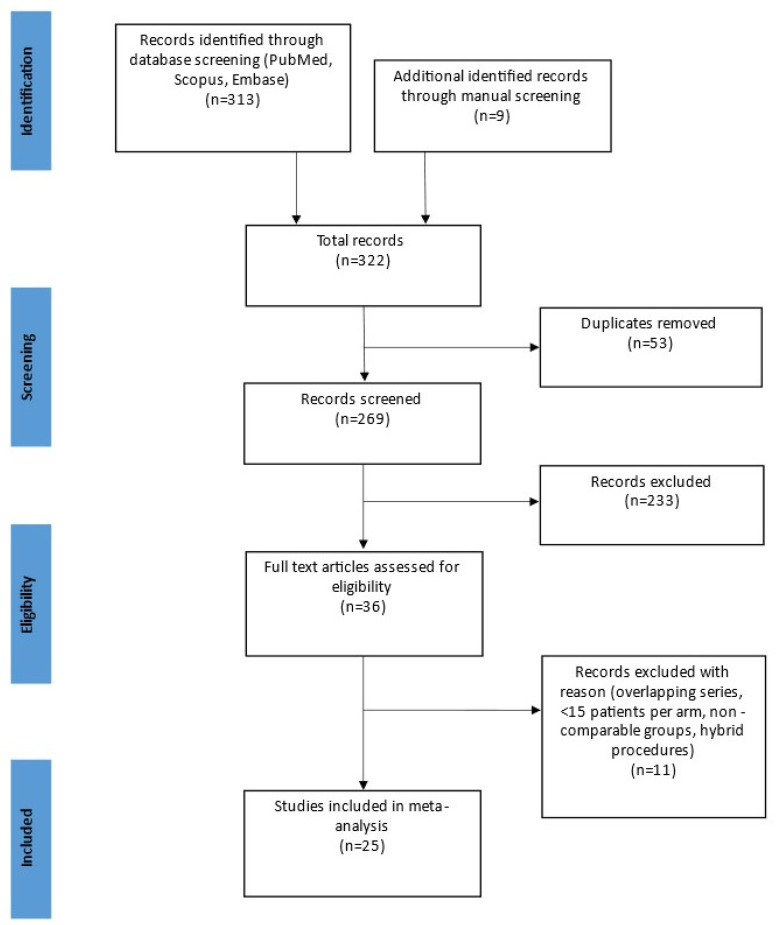
Flow-chart of included articles in accordance with the PRISMA guidelines [37] for systematic reviews and meta-analysis.

**Table 1 jcm-11-02387-t001:** Study details and quality assessment.

First Author	Study Type	Institution City, Country	Study Period	Study Design	LRC (n)	RRC (n)	Length of Follow-Up (d)	Quality Assessment	Reference
Yozgatli	Multicenter	Istanbul, Turkey	2015–2017	Retrospective	61	35	480/450	15	[15]
Ferri	Single	Madrid, Spain	2013–2017	Retrospective	35	35	1825/1825	15	[13]
Park	Single	Daegu, South Korea	2009–2011	Randomized	35	35	1825/1825	4	[7,11]
Spinoglio	Single	Milan, Italy	2005–2015	Retrospective	100	100	1825/1825	19	[8]
Migliore	Single	Cuneo, Italy	2010–2018	Retrospective	170	46	30/30	16	[14]
Hannan	Single	Limerick, Ireland	2017–2020	Retrospective	35	35	30/30	13	[17]
Tagliabue	Single	Lecco, Italy	2014–2019	Retrospective	68	55	180/180	17	[18]
Dohrn	Multicenter	Herlev, Denmark	2015–2018	Retrospective	3621	381	90/90	19	[10]
Merola	Multicenter	Naples, Italy	2012–2017	Retrospective	94	94	180/180	18	[16]
Ahmadi	Multicenter	Tweed Heads, Australia	2015–2018	Retrospective	42	59	n/a	15	[36]
Ngu	Single	Singapore, Singapore	2015–2017	Retrospective	16	16	30/30	14	[20]
Sorgato	Multicenter	Padoa, Italy	2018–2019	Retrospective	40	48	30/30	16	[21]
Widmar	Single	New York, USA	2009–2014	Retrospective	207	69	500/500	16	[19]
Gerbaud	Single	Paris, France	2013–2019	Retrospective	59	42	n/a	15	[22]
Mégevand	Single	Milan, Italy	2010–2015	Retrospective	50	50	30/30	17	[23]
Trastulli	Multicenter	Terni, Italy	2005–2014	Retrospective	134	102	30/30	15	[24]
Ceccarelli	Single	Foligno, Italy	2014–2019	Retrospective	29	26	30/30	15	[25]
De Angelis	Single	Paris, France	2012–2015	Retrospective	50	30	90/90	17	[26]
Deutsch	Single	Roslyn, USA	2004–2009	Retrospective	47	18	n/a	16	[27]
Haskins	Multicenter	Washington, USA	2012–2014	Retrospective	2405	89	30/30	15	[28]
Rawlings	Single	Peoria, USA	2002–2005	Prospective	15	17	n/a	15	[30]
deSouza	Single	Chicago, USA	2005–2009	Retrospective	135	40	n/a	16	[31]
Casillas	Single	Ann Arbor, USA	2005–2012	Prospective	110	52	n/a	16	[32]
Kang	Single	Seoul, South Korea	2007–2011	Retrospective	43	20	1200/1200	18	[33]
Dolejs	Multicenter	Indianapolis, USA	2012–2014	Retrospective	6521	259	n/a	16	[34]
Lujan	Single	Jackson, USA	2009–2015	Retrospective	135	89	n/a	17	[35]

Values are given in absolute numbers. Abbreviations: n/a = not available, d = days.

**Table 2 jcm-11-02387-t002:** Patients’ characteristics in the retrieved studies.

Author	Year	Age Mean (±SD)				Sex (*n*)						BMI (kg/m^2^)		ASA ≥2		Neoplasm	
		LRC		RRC		LRC			RRC			LRC	RRC	LRC	RRC	LRC	RRC
		Mean	SD	Mean	SD	f	m	Total	f	m	Total			*n* (%)	*n* (%)	*n* (%)	*n* (%)
Yozgatli [15]	2019	65	13	65	13	30	31	61	15	20	35	27	29	n/a	n/a	61(100)	35(100)
Ferri [13]	2021	68	n/a	70	n/a	15	20	35	12	23	35	25	23	31(89)	27(77)	32(91)	32(91)
Park [7,11]	2019	66.5	10.5	62.8	11.4	19	16	35	21	14	35	23.8	24.4	14(40)	20(57)	35(100)	35(100)
Spinoglio [8]	2018	71.2	10.6	71.2	10.2	54	46	100	44	56	100	25.8	25.1	91(91)	88(88)	100(100)	100(100)
Migliore [14]	2021	71.92	10.1	68.7	9.2	74	96	170	24	22	46	25.52	26.05	153(90)	35(76)	163(96)	43(93)
Hannan [17]	2021	69.7	n/a	66.5	n/a	17	18	35	17	18	35	n/a	n/a	21(60)	28(80)	28(80)	20(57)
Tagliabue [18]	2020	72	n/a	72	n/a	28	40	68	23	32	55	24.81	24.31	56(82)	42(76)	55(81)	41(75)
Dohrn [10]	2021	73	n/a	73	n/a	2000	1621	3621	196	185	381	25.7	25.6	2956(82)	295(77)	3616(100)	381(100)
Merola [16]	2020	72.09	9.5	69.4	10.3	33	61	94	34	60	94	27.97	26.94	83(88)	87(93)	94(100)	94(100)
Ahmadi [36]	2021	75	12	75	13	22	20	42	29	30	59	27	27	n/a	n/a	35(83)	43(73)
Ngu [20]	2018	69.6	9.6	68.6	10.9	10	6	16	6	10	16	24.7	23.7	16(100)	16(100)	16(100)	15(94)
Sorgato [21]	2021	68	10	71	12.2	12	28	40	21	27	48	26.6	25.6	37(93)	46(96)	38(95)	41(85)
Widmar [19]	2016	64	n/a	66	n/a	122	85	207	36	33	69	n/a	n/a	n/a	n/a	n/a	n/a
Gerbaud [22]	2019	72	8.6	67	8.6	28	31	59	21	21	42	24	26	n/a	n/a	37(63)	30(71)
Mégevand [23]	2019	69.6	n/a	70.3	n/a	26	24	50	22	28	50	25.25	26.2	43(86)	44(88)	35(70)	41(82)
Trastulli [24]	2015	71.01	n/a	71.2	11.6	57	77	134	46	56	102	25.76	25.6	122(91)	94(92)	113(84)	81(79)
Ceccarelli [25]	2021	75	11.7	69.1	9.4	14	15	29	6	20	26	24.2	24.4	20(69)	24(92)	24(83)	24(92)
De Angelis [26]	2016	71.1	12.9	71	8.5	31	19	50	15	15	30	25.26	26.43	46(92)	30(100)	50(100)	30(100)
Deutsch [27]	2012	70.8	14.6	65.2	12	22	25	47	6	12	18	28	25	24(96)	4(22)	28(60)	18(100)
Haskins [28]	2018	68.3	12.6	68.9	11.8	1279	1126	2405	40	49	89	28.5	29.3	2363(98)	89(100)	2405(100)	89(100)
Rawlings [30]	2007	63.1	17.5	64.6	11.7	9	6	15	9	8	17	28.3	25.7	n/a	n/a	6(40)	2(12)
deSouza [31]	2010	65.32	18.7	71.4	14.1	73	62	135	18	22	40	26.57	27.33	118(87)	35(88)	66(49)	18(54)
Casillas [32]	2014	71	12	65	12	41	69	110	27	25	52	27	26.9	108(98)	51(98)	110(100)	52(100)
Kang [33]	2016	65.7	13.2	66	9.6	21	22	43	11	9	20	23	23.5	22(51)	9(45)	43(100)	20(100)
Dolejs [34]	2017	n/a	n/a	n/a	n/a	2913	3608	6521	133	126	259	n/a	n/a	n/a	n/a	3247(50)	116(45)
Lujan [35]	2018	72.6	11.4	70.9	9.6	74	61	135	41	48	89	27.1	27.8	130(96)	88(99)	n/a	n/a

Values are given in absolute numbers or percentage. Abbreviations: n/a = not available, SD = Standard deviation, f = female, m = male, BMI = body mass index, ASA = American Society of Anesthesiologists. Gray: Laparoscopic group, white: Robotic group.

**Table 3 jcm-11-02387-t003:** Meta-analysis of perioperative outcome of included studies.

Variable	LRC	RRC	OR/MD	*p*-Value	I^2^	References
Age, years	70.27 ± 3.00	68.79 ± 2.90	1.48 (0.11–2.84)	0.03	47%	[3,7,8,11,15,16,20,21,22,25,26,27,28,30,31,32,35,36]
Neoplasm, *n*	7539/11,017	946/1229	1.22 (0.91–1.64)	0.17	27%	[10,13,14,17,18,20,21,22,23,24,25,27,30,31,34,36]
Operative time (min)	165.31 ± 43.08	207.38 ± 189.13	−42.01 (−51.06–32.96)	<0.001	89%	[7,8,11,13,14,16,18,19,20,21,22,23,24,25,26,27,28,30,31,33,34,36]
Blood loss (mL)	63.57 ± 35.21	53.62 ± 34.02	10.03 (1.61–18.45)	0.02	65%	[7,10,11,15,22,24,26,27,30,31,33,35]
Conversion, *n*	1155/11629	94/1534	1.53 (1.08–2.17)	0.02	14%	[8,10,13,14,16,17,18,19,22,23,24,26,27,30,31,32,33,34,35]
Intracorporeal Anastomosis, *n*	329/4308	468/860	0.03 (0.00–0.20)	<0.001	90%	[7,10,11,13,19,21,22,35,36,43]
Time to first flatus (d)	2.46 ± 2.14	2.30 ± 2.08	0.15 (−0.18–0.48)	0.38	93%	[7,8,11,13,14,15,16,18,20,23,24,26,27,33]
Mortality, *n*	126/13,388	18/1198	0.66 (0.41–1.06)	0.08	0%	[8,10,14,16,26,27,28,31,32,34,35]
Overall Morbidity, *n*	3093/14,242	464/1825	1.01 (0.86–1.19)	0.88	22%	[7,8,10,11,13,14,15,16,17,18,19,20,21,22,23,24,25,26,27,28,31,32,33,34,35,36]
Non-surgical complications, *n*	693/13,515	119/1406	0.93 (0.70–1.23)	0.6	9%	[8,10,15,18,21,22,23,24,25,26,27,28,31,32,33,34,36]
Incisional hernia, *n*	53/389	12/176	1.51 (0.78–2.95)	0.22	0%	[19,27,35]
Postoperative hemorrhage, *n*	573/10,013	55/1178	0.88 (0.64–1.21)	0.43	0%	[7,8,11,15,16,18,21,22,23,24,25,27,28,30,31,33,34,35,36]
Postoperative ileus, *n*	962/10,257	70/1209	1.30 (0.91–1.87)	0.14	18%	[7,8,11,15,18,19,21,22,23,24,26,27,28,30,31,32,33,34,35,36]
Wound infection, *n*	618/10,074	60/1076	1.15 (0.84–1.57)	0.39	0%	[7,8,11,15,17,18,19,21,22,24,25,28,31,32,34,35]
Anastomotic leakage, *n*	273/11,552	34/1557	1.02 (0.69–1.50)	0.94	0%	[7,8,10,11,14,15,16,17,18,19,21,22,23,24,26,27,30,32,34,35]
Abdominal abscess, *n*	13/966	10/526	0.75 (0.34–1.64)	0.47	0%	[7,11,14,15,18,19,20,21,23,24,26,31]
Hospital stay (d)	6.15 ± 31.77	5.31 ± 1.65	0.84 (0.29–1.38)	0.003	87%	[7,8,11,14,16,18,20,21,22,23,24,25,26,27,28,30,31,33,34,35]

Values are given in mean ± SD or in absolute numbers. Abbreviations: OR: Odds ratio, MD: Mean difference, d: days, mL: milliliter, min: minutes.

**Table 4 jcm-11-02387-t004:** Meta-analysis of oncological outcome.

Variable	LRC	RRC	OR/MD	*p*-Value	I^2^	References
Lymph nodes harvested	22.97 ± 5.94	23.82 ± 6.76	−0.85 (−2.19–0.48)	0.21	75%	[7,8,10,11,14,15,16,18,20,21,22,23,25,28,33,35,36,43]
Disease free survival (5 years)	178/213	162/190	0.87 (0.50–1.51)	0.62	0%	[7,8,11,13,33]
Overall survival (5 years)	172/213	157/190	0.90 (0.54–1.52)	0.7	0%	[7,8,11,13,33]

Values are given in mean ± SD or in absolute numbers. Abbreviations: OR: Odds ratio, MD: Mean difference.

## Data Availability

The datasets generated and/or analyzed during the current study are available from the corresponding author on reasonable request.

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
