# Peer review of "Short- and Long-Term Outcome of Laparoscopic- versus Robotic-Assisted Right Colectomy: A Systematic Review and Meta-Analysis"

_jcm, 2022, doi:10.3390/jcm11092387_

Round 1
Reviewer 1 Report
The authors performed a systematic review and meta-analysis to compare short and long-term outcomes of laparoscopic vs. robotic-assisted right colectomy. With about more than 16000 patients analyzed, the results were that whereas operative time and surgical and total costs are higher in the robotic group, blood loss, conversion rate to open procedure and hospital stay were lower in the robotic group. Postoperative mortality and morbidity as well as oncological outcomes did not differ between groups.
This article is interesting, well written and structured. No grammatical adjustments are needed. I have some comments about your work:
Main comments:
- Did you consider registering your meta-analysis and systematic review with PROSPERO? This would prevent from duplicated data and publications.
- In the methodology you talk about cost analysis and in the discussion you affirm that costs (surgical specific- and total costs) were significantly higher in the robotic group, but there are no references for these findings in the “Results” paragraph. You should present your data in “Results” also.
Minor comments:
- From 16099 patients, only 1842 (11%) underwent robotic right colectomy. Do you think this could involve some bias risk? Do you think that with more patients and maybe more learning curve, operative times could be lower in the robotic group? Consider to include this aspect in the discussion.
- You describe that age was higher in the LRC group than in the RRC group. Do you think this could affect postoperative outcomes, such as length of hospital stay? Could this be a confusion factor for length of hospital stay, which is shorter in the RRC group? Could you please clarify this aspect?
- You also describe that there are more intracorporeal anastomosis performed in the RRC group than in the LRC group. In several papers, intracorporeal anastomosis has been related with a shorter length of hospital stay, quicker recovery of bowel function and less blood loss. This could also be a confusion factor. Could you please clarify this aspect as well?
- When analyzing the oncological findings, in paragraph 3.4 you describe only 4 studies performing long term observation of the patients, but you include 5 of them in the references (7, 8, 11, 13, 33). In the discussion, you only include the references of 4 (7, 8, 13, 33). This should be fixed.
Author Response
Thank you for reviewing this manuscript and for your comments!
- Did you consider registering your meta-analysis and systematic review with PROSPERO? This would prevent from duplicated data and publications.
We did not consider a PROSPERO registration in the past. A completed review can not be registered any more.
- In the methodology you talk about cost analysis and in the discussion you affirm that costs (surgical specific- and total costs) were significantly higher in the robotic group, but there are no references for these findings in the “Results” paragraph. You should present your data in “Results” also.
We also performed a cost analysis: We deleted this section because of journals’ aims and scopes. If we should add it, please do not hesitate to contact me. We deleted “costs” in the methods-section and discussion section as well.
|
Surgery costs (€) |
3,900 ± 1,677 |
8,156 ± 0,458 |
-4,16 [-7,12 - -1,21]* |
0,006 |
93% |
|
Total costs (€) |
7,647 ± 1,307 |
10,306 ± 1,507 |
-2,66 [-5,17 - -0,15]* |
0,04 |
96% |
- From 16099 patients, only 1842 (11%) underwent robotic right colectomy. Do you think this could involve some bias risk? Do you think that with more patients and maybe more learning curve, operative times could be lower in the robotic group? Consider to include this aspect in the discussion.
We have added this very interesting commentary in the discussion section (P12, L207).
- You describe that age was higher in the LRC group than in the RRC group. Do you think this could affect postoperative outcomes, such as length of hospital stay? Could this be a confusion factor for length of hospital stay, which is shorter in the RRC group? Could you please clarify this aspect?
We added age in Table 3 and added a sentence into the discussion section (P12, L182). You are right, especially in retrospective studies younger and healthier patients were selected for robotic procedures. This could have an influence on hospital stay.
- You also describe that there are more intracorporeal anastomosis performed in the RRC group than in the LRC group. In several papers, intracorporeal anastomosis has been related with a shorter length of hospital stay, quicker recovery of bowel function and less blood loss. This could also be a confusion factor. Could you please clarify this aspect as well?
We tried to clarify in the discussion section (P12, L196).
- When analyzing the oncological findings, in paragraph 3.4 you describe only 4 studies performing long term observation of the patients, but you include 5 of them in the references (7, 8, 11, 13, 33). In the discussion, you only include the references of 4 (7, 8, 13, 33). This should be fixed.
You are right. Studies 8 and 11 are from Park et al. 11 was published as a follow-up of the study published from Park in 2012. In 2018 he published a follow-up including oncological long-term data. We cancelled Nr. 8 in the table because 8 does not include long term data.
Reviewer 2 Report
The authors systematically reviewed short and long term outcomes after right colectomy comparing laparoscopic and robotic-assisted approaches. Given the growing availability and diffusion of robotic instrumentation, this topic is of increased interest in the scientific community. To date, no guidelines or strong evidence are available to define indications for a laparoscopic rather than robotic approach for right colon resections. In this systematic review, the search strategy sounds adequate, and the number of patients appropriate. I have several comments for the Authors.
- The long-term oncological outcomes are among the primary endpoints which have never been previously assessed in a metanalysis. However, no reference to surgical indication, pathological diagnosis, (neo)adjuvant treatments, or tumour staging is reported in the present analysis. Could this information be retrieved? Could the surgical approach (robot-assisted vs laparoscopic) depend on the complexity/stage of the colonic lesion?
- One of the most critical issues about robotic-assisted surgery is costs. This aspect was not considered among the outcomes of this analysis. It would be useful to evaluate this aspect as well. Do a slight decrease in blood loss (median 10 ml) and length of hospital stay (median 1 day) justify the higher expenses in all right-colonic resections?
- The Authors highlighted a significantly higher number of intracorporeal anastomoses in the robotic technique than in the laparoscopic. The robotic approach is also associated with longer operative times. Might the Authors compare the two approaches by the operative time, considering the intra/extracorporeal anastomoses separately?
- The results section could be implemented. Comments are needed on the heterogeneity of the included studies at least for primary endpoints.
Author Response
Dear Reviewers, dear Editors!
Thank you very much for giving us the opportunity to resubmit a revision of the manuscript. All corrections are highlighted (marked in yellow). If you have further requests, please do not hesitate to contact me!
Thanks in advance!
Sincerely
Peter Tschann
Reviewer 2:
Thank you for reviewing this manuscript and for your commentaries.
- The long-term oncological outcomes are among the primary endpoints which have never been previously assessed in a metanalysis. However, no reference to surgical indication, pathological diagnosis, (neo)adjuvant treatments, or tumour staging is reported in the present analysis. Could this information be retrieved? Could the surgical approach (robot-assisted vs laparoscopic) depend on the complexity/stage of the colonic lesion?
We tried to clarify. Most of the studies did not include pathological staging. To compare both groups regarding the indication/complexity was not possible, because most of the studies were of retrospective design. We added this in the limitation section (P12,L210)
- One of the most critical issues about robotic-assisted surgery is costs. This aspect was not considered among the outcomes of this analysis. It would be useful to evaluate this aspect as well. Do a slight decrease in blood loss (median 10 ml) and length of hospital stay (median 1 day) justify the higher expenses in all right-colonic resections?
|
Surgery costs (€) |
3,900 ± 1,677 |
8,156 ± 0,458 |
-4,16 [-7,12 - -1,21]* |
0,006 |
93% |
|
Total costs (€) |
7,647 ± 1,307 |
10,306 ± 1,507 |
-2,66 [-5,17 - -0,15]* |
0,04 |
96% |
We made a cost analysis. We deleted it from this manuscript because of aims and scopes of the journal.
- The Authors highlighted a significantly higher number of intracorporeal anastomoses in the robotic technique than in the laparoscopic. The robotic approach is also associated with longer operative times. Might the Authors compare the two approaches by the operative time, considering the intra/extracorporeal anastomoses separately?
This is an interesting point. We do not think that type of anastomosis is influencing operative time significantly. Furthermore, in consideration on the fact, that in robotic surgery only experienced colorectal surgeons are performing the majority of the cases the time to perform an anastomosis – independent of type – is not that long. Moreover, docking time, changing of instruments is more time consuming in robotic surgery than in conventional laparoscopy. We discussed this interesting point in P11, L165 (highlighted).
- The results section could be implemented. Comments are needed on the heterogeneity of the included studies at least for primary endpoints.
Data heterogeneity of included studies was discussed in P12, L205 (highlighted).
Round 2
Reviewer 2 Report
Dear Authors,
Thank you for your prompt response.
I read all of your commentaries; I had some issues.
- Considering oncological outcomes, it is highly relevant to report at least the inclusion and exclusion criteria adopted in each included study, like preoperative staging inclusion criteria, emergency vs election, and possibly access to adjuvant therapy between groups. I find it hard to believe those parameters are not reported. The limited number of studies makes this analysis difficult, but having chosen oncological outcomes as a primary endpoint and remarking several times how this is the innovative element of this meta-analysis; thus it requires deeper and more specific comments on it.
- Regarding surgical costs, the table seems to be important in defining the accessibility of robotic-assisted surgery; however, the table is not readable, considering lacking titles and legends. I presume both p values are significant (are they a “p”?). I suggest you add this analysis to the results or at least as a supplementary.
- Percentages should be added in table 2 to improve clarity.
Author Response
Dear Reviewers, dear Editors!
Thank you very much for giving us the opportunity to resubmit a revision of the manuscript. All corrections are highlighted (marked in green). If you have further requests, please do not hesitate to contact me!
Addition supplement material was added. Costs were included as separate supplement table and added into the discussion. In respect to Journals’ aims and scopes, I took this compromise and left it out from the main Table.
Thanks in advance!
Sincerely
Peter Tschann
Reviewer 2:
Thank you for reviewing the revised manuscript and for your commentaries.
- Considering oncological outcomes, it is highly relevant to report at least the inclusion and exclusion criteria adopted in each included study, like preoperative staging inclusion criteria, emergency vs election, and possibly access to adjuvant therapy between groups. I find it hard to believe those parameters are not reported. The limited number of studies makes this analysis difficult, but having chosen oncological outcomes as a primary endpoint and remarking several times how this is the innovative element of this meta-analysis; thus it requires deeper and more specific comments on it.
We carefully reviewed all included studies, especially the studies with a long-term follow-up. Supplement Table 1 shows TNM or UICC stage of the studies with a long-term follow-up. Additionally, those studies were screened for adjuvant therapy. Only one study was reporting about adjuvant treatment (Park et al.- added in Supplement Table 1). All other studies showed no detailed information about adjuvant treatment. Supplement Table 2 shows TNM (A) or UICC (B) stage of all studies which were included into this meta-analysis. There was no difference between both groups regarding oncological stage. Oncological stage was neither influencing the decision for conventional laparoscopy nor for robotic surgery. In addition, you are completely right that the limited number of studies with a long-term follow-up should be mentioned. I added a critical view regarding the interpretation of those data into the discussion section (P11,P12, L180-188, P12, L218). Additionally, percentage of ASA and of included cancer cases are added in Table 2 for a clearer understanding of indication of included studies.
- Regarding surgical costs, the table seems to be important in defining the accessibility of robotic-assisted surgery; however, the table is not readable, considering lacking titles and legends. I presume both p values are significant (are they a “p”?). I suggest you add this analysis to the results or at least as a supplementary.
We added the meta-analysis of costs in Supplement Table 3 and as text in the result section (P11, L148-151). We added one sentence in the discussion section (P11, L167).
- Percentages should be added in table 2 to improve clarity.
Added into Table 2 (ASA and Number of cancer cases =”Neoplasm”).
